# Language Quantized AutoEncoders:
# Towards Unsupervised Text-Image Alignment

**Hao Liu**
UC Berkeley
hao.liu@cs.berkeley.edu

**Wilson Yan**
UC Berkeley
wilson1.yan@berkeley.edu

**Pieter Abbeel**
UC Berkeley
pabbeel@cs.berkeley.edu

## Abstract

Recent progress in scaling up large language models has shown impressive capabilities in performing few-shot learning across a wide range of natural language tasks. However, a key limitation is that these language models fundamentally lack grounding to visual perception - a crucial attribute needed to extend to real world tasks such as in visual-question answering and robotics. While prior works have largely connected image to text through pretraining or fine-tuning, learning such alignments are generally costly due to a combination of curating massive datasets and large computational burdens. In order to resolve these limitations, we propose a simple yet effective approach called **L**anguage-**Q**uantized **A**uto**E**ncoder (LQAE), a modification of VQ-VAE that learns to align text-image data in an *unsupervised* manner by leveraging pretrained language model denoisers (*e.g.*BERT). Our main idea is to encode images as sequences of text tokens by directly quantizing image embeddings using a pretrained language codebook. We then feed a masked version of the quantized embeddings into a BERT to reconstruct the original input. By doing so, LQAE learns to represent similar images with similar clusters of text tokens, thereby aligning these two modalities *without* the use of aligned text-image pairs. We show LQAE learns text-aligned image tokens that enable few-shot multi-modal learning with large language models, outperforming baseline methods in tasks such as image classification and VQA while requiring as few as 1-10 image-text pairs[1].

## 1 Introduction

Large language models powered by transformers [25] have shown impressive capabilities in modeling natural language [see e.g. 2, 17, 6, 30], demonstrated through their abilities to quickly learn novel tasks such as question answering, chatbots, and machine translation from just a few examples without finetuning. This so called few-shot learning turns out to be competitive with conventional task specific methods in various NLP tasks and is being rapidly adapted to more new tasks and styles of generations.

Despite these impressive capabilities, a key limitation of such large language models is that they cannot 'see' the visual world. Being able to 'see' the world is crucial for many real world applications where processing abundant complex sensory data is a must, such as robotics, visual question answering, and grounding. Such a limitation severely hinders further applicability of large transformer models to a variety of downstream tasks.

Driven by these impressive results for large language models, much of recent work has started to bring text and image together, and leverage these aligned modalities to perform a variety of applications, such as text to image generation [29, 3], open ended classification [23, 18, 1], and image

---

[1]Code: `https://github.com/lhao499/language-quantized-autoencoders`

37th Conference on Neural Information Processing Systems (NeurIPS 2023).

editing [9, 15, 19]. Frozen [23] trains a visual encoder to adapt images to pretrained language model for predicting aligned text, and demonstrates few-shot learning in classification and VQA.

However, these works generally require large amounts of aligned data - Frozen is pretrained on Conceptual Captions [22] (3M text-image pairs), and many prior methods use CLIP [18] trained on 300M text-image pairs. Collecting and curating such as large amount aligned data can be expensive and costly, and substantially difficult in arbitrary pairs of modalities. On the other hand, it is comparably easier to aggregate data within single modalities, such as scraping text and video data independently rather than curating aligned pairs. In this case, the primary difficulty in leveraging these cross-modal applications requires us to learn some implicit alignment between these two modalities.

To resolve this issue in text-image learning, we propose **L**anguage **Q**uantized **A**uto**E**ncoder (LQAE) to encode unaligned images to language space without relying on paired text-image data. In order to do so, we adopt the vector-quantization approach introduced in VQ-VAE [24] and use a frozen language model codebook to replace the standard learned codebook. Specifically, we propose to incorporate a pretrained language denoiser (*e.g.* BERT) into the VQVAE pipeline by randomly masking the encoded codes (quantized to correspond to natural language tokens) to feed into BERT, and then reconstructing the original image from the output BERT features. Intuitively, the reconstruction objective from BERT features encourages the encoder to encode images that are more easily denoised by BERT, which generally align with more natural language semantics. Since we train the encoder and decoder to minimize image reconstruction error while keeping BERT-like model fixed, similar images must be mapped to similar text tokens in order to minimize reconstruction loss, though with no guarantee for the learned alignment to correspond with human interpretation (*e.g.* the word "dog" matching with pictures of dogs).

In our experiments, we first show that it is possible to train an auto-encoder that uses a language embedding space. Then, we carefully investigate the quality of the textual representations of images through three evaluations: (i) few-shot classification (Open-Ended miniImageNet) (see Section 4.2); (ii) visual-question answering (Fast VQA) (see Section 4.3); (iii) linear classification experiments (see Section 4.4). Our findings indicate that the textual semantics are effectively retained, allowing for strong performance on these tasks. Finally, our ablation study shows that using large language models (*e.g.*, GPT-3 Davinci) improves results and masking a high mask ratio is crucial for learning textual representations of images for text-image understanding.

Our contributions are:

- We propose LQAE, a method for learning aligned textual representations of images by leveraging pretrained language models.

- We show that LQAE interface allows using large language models for few-shot image classification through standard prompting without any need for finetuning. We demonstrate that LQAE outperforms existing SOTA methods on few-shot learning while requiring orders of magnitudes fewer text-image pairs and computational resources.

- We show that LQAE allows using BERT for image linear classification. We also conduct an ablation study showing that masking a high mask ratio in training language quantization is crucial for learning textual representations of image for text-image understanding.

## 2 Method

In this work, we introduce the **L**anguage-**Q**uantized **A**uto**E**ncoder (LQAE), a modification of VQ-VAE that learns to align text-image data in a data efficient manner by leveraging off-the-shelf language denoisers such as RoBERTa [14]. The overall architecture of our framework is shown in Figure 1.

### 2.1 VQ-VAE

VQ-VAE is an autoencoder that learns to compress image data into a set of discrete latents. The model consists of an encoder $E$, decoder $D$, and codebook $C$. Encoder $E$ encodes image $x \in \mathbb{R}^{H \times W \times 3}$ to produce $E(x) = h \in \mathbb{H}' \times \mathbb{W}' \times \mathbb{D}$, which is quantized by codebook $z = C(h)$ through nearest neighbors lookup. The quantized codes are then fed to the decoder ($\hat{x} = D(z)$) to reconstruct the

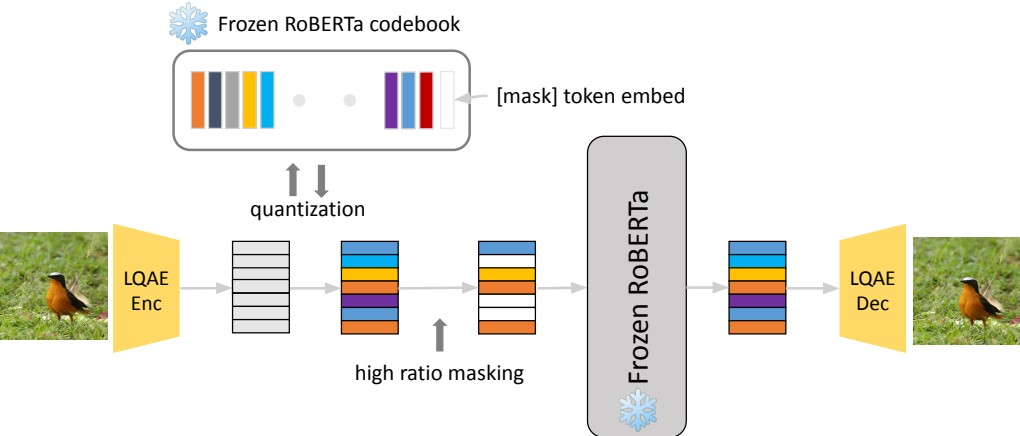

Figure 1: Model architecture of Language Quantization AutoEncoder (LQAE). Images are encoded to a sequence of embeddings, then vector quantized using RoBERTa codebook, followed by a high ratio masking and a frozen RoBERTa prediction, and finally decoded to reconstruct the original image. All parameters are frozen except the LQAE encoder and decoder

original image $x$, and optimizes the following loss:

$$\mathcal{L} = \| x - \hat{x} \|_2^2 + \| \operatorname{sg}(h) - z \|_2^2 + \beta \| h - \operatorname{sg}(z) \|_2^2 \tag{1}$$

consisting of an $\ell_2$ reconstruction loss, codebook loss, and commitment loss. A straight-through estimator is used in order for gradient to flow through the quantization step. We use ViT-base [8] as image encoder and decoder.

## 2.2 Learning Text-Image Alignment

In order to learn text-aligned image representations, we propose several key modifications on the original VQ-VAE architecture to learn connected text-image discrete representations by incorporating a pretrained language denoiser (*e.g.* any BERT-like model).

**Pretrained codebook:** First, we replace the learned codebook $C$ with a *fixed* codebook from our pretrained language model. The codebook remains frozen throughout training, so there is no need for the standard VQ codebook loss $\| \operatorname{sg}(h) - z \|_2^2$. This way, the encoder learns to directly map images into a latent space defined by text token ids, where resulting discrete encodings can be directly rendered into text.

**Incorporating pretrained language denoisers:** Although replacing the VQ codebook with a pretrained language model codebook allows for our method to directly encode images into texual representations, there is less guarantee that the resulting encoded text will align well with natural language semantics. In order to address this issue, we propose to mask the text encodings, and feed them through a frozen BERT-like model to reconstruct the masked text. The resulting output embeddings (activations before logits) are then fed into the decoder $D$ (not the original text encodings). This way, encoder $E$ is encouraged to learn text encodings that align better with standard text so that the BERT model can more easily reconstruct the original full encoding to feed into the decoder. In addition, we found that adding a low-weighted standard BERT loss helped in downstream performance. The final loss is written as

$$\mathcal{L} = \| x - \hat{x} \|_2^2 + \beta \| h - \operatorname{sg}(z) \|_2^2 + \alpha \log p(z \mid z_m),$$

where $\alpha$ and $\beta$ are hyperparameter and sg refers to the stop gradient. $\alpha = 0.001$ and $\beta = 0.005$ are are determined through a hyperparameter sweep and used as default unless otherwise mentioned. $z_m$ is the masked version of input $z$ fed into the BERT model.

We remark that since LQAE learns to encode image to text and text to image without using aligned image-text supervision, LQAE does not need to generate human interpretable text representations,

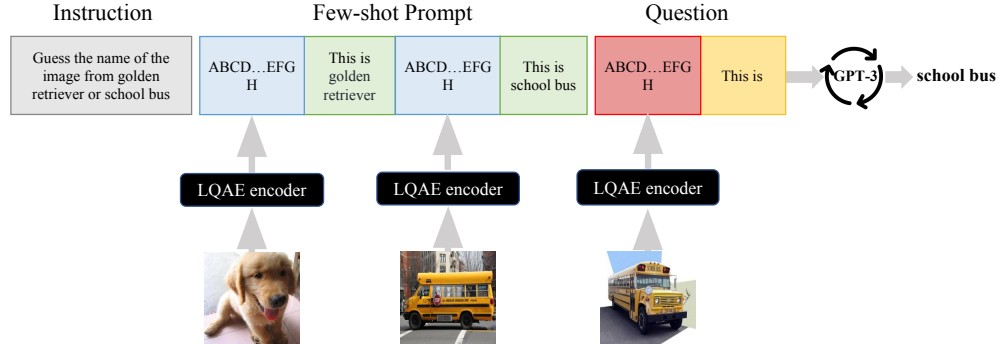

Figure 2: Language Quantization AutoEncoder (LQAE) can be used for few-shot image classification and visual question answering by leveraging the in-context learning ability of large language models *e.g.*, GPT-3

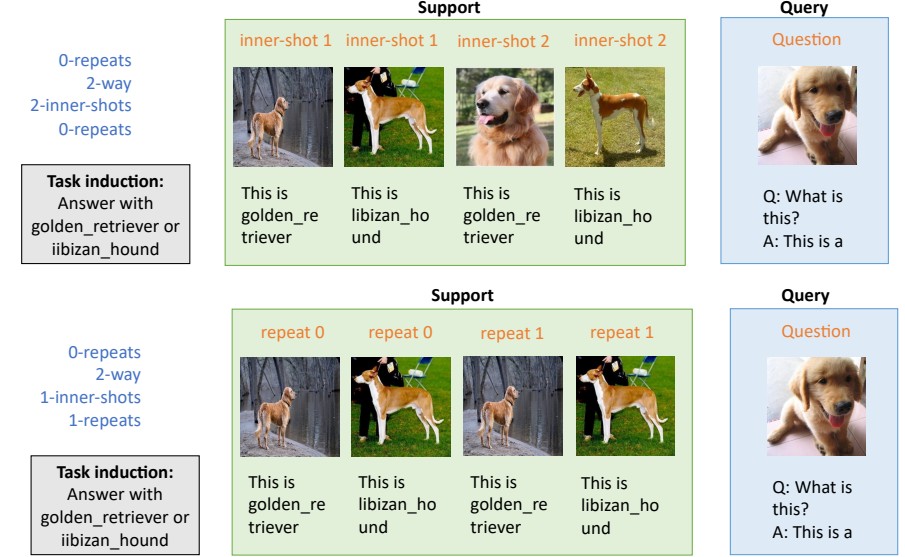

Figure 3: Visual examples of the terminologies used in our few-shot image classification and visual question answering experiments.

*i.e.*, an image of dog can have text representation describing something totally unrelated such as rivers. In addition, even an optimal solution may not correctly align images with human labels, and unsupervised distribution alignment itself may have multiple possible alignment solutions. However in our experiments, we show that the resulting learned textual representations of images provide meaningful visual features that are interpretable when few-shot prompted with a large language model.

**Inference.** At test time, LQAE provides a simple interface for both supervised learning and open ended few-shot learning. As shown in Figure 2, we can use LQAE to encode each image into a sequence of text tokens, and similar to Frozen, interleave image and text representations to form a few-shot prompt for a large language model such as GPT-3 or InstructGPT [2, 17] to classify a given test-time image. In our experiments, we show that by leveraging LQAE representations for few-shot prompting in LLMs, our method only requires a few paired examples for strong task performance.

## 3   Experimental Setup

**Few-shot Classification.** We condition LQAE on a sequence of interleaved images and text to evaluate the model's ability at 'inducing' the task to the model in order to improve its performance.

Following prior work, we define the following terminology used in our settings across all tasks. Figure 3 gives a visual illustration of these concepts.

- Task induction: An introductory text that provides information about the task to the model using natural language. This text appears before the sequence of images and encoded text and is used to explain what the model is expected to do, for instance, "Please answer the question"

- Number of ways: This refers to the total number of categories involved in the task, for example, the distinction between dogs and cats.

- Number of inner-shots: This refers to the number of unique examples of each category that are presented to the model, such as the number of different images of dogs. In prior studies using Mini-Imagenet, the unique examples of each category were also referred to as *shots*.

- Number of repeats: The "number of repeats" specifies the number of times each unique example of a category is repeated in the context presented to the model. This setting is used to study the model's ability to integrate visual information about a category through an evaluation technique known as ablation following prior work [23].

**Linear Classification.** For LQAE, we use features from intermediate RoBERTa layers for image representations. When comparing against VQ-VAE features, we replicate the feature dimensions by the number of RoBERTa layers to match the number of linear classification parameters used in our method.

## 4 Main Results

### 4.1 Training Details

We train our LQAE on the ImageNet dataset, and use RoBERTa-base[2] as our pretrained language denoising model.

We operate on $256 \times 256$ images at both train and test-time; images that are not square are first resized to $256 \times 256$. ViT encoder and decoder patch size is $16 \times 16$. Adam [11] optimizer is used for training with peak learning rate $1.5 \times 10^{-4}$ and weight decay 0.0005. Training takes 100 epochs with 5 warmup epochs. Batch size is 512 and training is distributed between 128 TPU-v3 on Google Cloud.

### 4.2 Evaluation on Fewshot Classification

To quantify few-shot performance, we evaluate our method on the Real-Name Open-Ended miniImageNet defined in Tsimpoukelli et al. [23], where a model is few-shot prompted with a few examples of images per class, and asked to classify a new image. We compare our method against several baselines, which can be divided into several distinct categories:

- **No image pretraining**: Our ASCII baseline constructs text representations for each image by converting them to $64 \times 64$ ASCII images. We do not use $256 \times 256$ resolution for this baseline is because the resulting few-shot ASCII codes are tens of thousands long that GPT-3 does not support.

- **Text-image pretraining**: Frozen requires pretraining a joint image-language on aligned text-image data, and uses embeddings from the pretrained visual encoder.

- **Image-only pretraining**: MAE + Linear uses a pretrained MAE on ImageNet and fits a linear classifier on each set of few shot examples to predict the given test image. Both MAE + Linear and LQAE *do not require any text-image aligned data during pretraining*, and rely solely on models trained in individual domains. At most 5 aligned pairs are provided in each test-time example to measure few-shot learning.

For all methods except **MAE + Linear**, we follow the same evaluation structure as Frozen by constructing the few-shot prompt through alternating text class label and visual representation tokens - embeddings in the case of Frozen, and visual text encodings for LQAE and ASCII. For LQAE and ASCII, we prompt OpenAI's text-davinci-003 model for few-shot prediction.

---

[2]available at `https://huggingface.co/roberta-base`

Table 1: Performance of LQAE and baselines on Open-Ended miniImageNet 2-Way benchmark. Randomly picking between the two class labels (then emitting the EOS token) would yield 50% accuracy.

| Few-shot Setting | Task Induction | ✗ | ✓ | ✓ | ✓ | ✓ | ✓ | ✓ | Avg |
|---|---|---|---|---|---|---|---|---|---|
| | Inner Shots | 1 | 1 | 3 | 5 | 1 | 1 | 1 | |
| | Repeats | 0 | 0 | 0 | 0 | 1 | 3 | 5 | |
| No image or text | ASCII (64x64 img) | 0 | 5.2 | 5.9 | 6.5 | 4.5 | 4.8 | 5.2 | 4.59 |
| Image pretrain + Image-text finetune | MAE + Linear | 0 | 8.9 | 11.4 | 13.5 | 12.8 | 15.6 | 19.8 | 11.71 |
| Image-text pretrain | Frozen | **1.7** | 33.7 | 66 | 66 | 63 | 65 | 63.7 | 51.3 |
| Image Pretrain | untrained LQAE | 0 | 8.2 | 13.8 | 14.5 | 10.4 | 12.7 | 15.6 | 10.74 |
| | LQAE (ours) | 1.5 | **35.2** | **68.2** | **69.8** | **68.5** | **68.7** | **65.9** | **53.97** |

Table 2: Performance of LQAE and baselines on Open-Ended miniImageNet 5-Way benchmark. Randomly picking between the two class labels (then emitting the EOS token) would yield 20% accuracy.

| Few-shot Setting | Task Induction | ✗ | ✓ | ✓ | ✓ | ✓ | ✓ | ✓ | Avg |
|---|---|---|---|---|---|---|---|---|---|
| | Inner Shots | 1 | 1 | 3 | 5 | 1 | 1 | 1 | |
| | Repeats | 0 | 0 | 0 | 0 | 1 | 3 | 5 | |
| No image or text | ASCII (64x64 img) | 0 | 0 | 0 | 0 | 0 | 0 | 0 | 0 |
| Image pretrain + Image-text finetune | MAE + Linear | 0.3 | 2 | 2.5 | 3.2 | 3.1 | 3.5 | 3.6 | 2.6 |
| Image-text pretrain | Frozen | 0.9 | 14.5 | 34.7 | 33.8 | **33.8** | 33.3 | 32.8 | 26.26 |
| Image Pretrain | untrained LQAE | 0 | 1.2 | 1.6 | 2.3 | 2.1 | 1.9 | 2.3 | 1.63 |
| | LQAE | **1** | **15.7** | **35.9** | **36.5** | 31.9 | **36.4** | **45.9** | **29.04** |

Tables 1 and 2 show results on 2-way and 5-way few-shot classification respectively. LQAE performs better than baselines across all evaluation settings, substantially outperforming all baselines that do not have access to text-image pairs. In addition, Frozen is able to benefit from text-image pretraining on Conceptual Captions, yet still performs worse than LQAE. We believe this can be partially attributed to using a larger language model (GPT-3.5) compared to Frozen. However, our method does not require any model fine-tuning, which would be prohibitively expensive to run a method such as Frozen on a GPT-3.5 model.

Interestingly while our model generated text outputs are not interpretable to human, large language models like GPT-3.5 can successfully do few-shot learning from them. This suggests LQAE and BERT-like model generated text tokens contain patterns that can be successfully captured and leveraged by powerful large language models. This finding is related to prior works that found few-shot learning more rely on formats similarity and patterns rather than the exact semantic meaning of prompts [16, 12, 27].

## 4.3 Evaluation on Visual Question Answering

To further evaluate LQAE, we use the Real-Fast-VQA dataset from Frozen [23], a challenging semantic task that aims to answer questions in a free-form manner based on an image, such as "what object is the person holding?" or "what color is the car?" Following the Frozen paper, we consider 2-way setting and assess model's performance using different numbers of inner-shots.

Table 3 shows the results. Trivial baselines such as ASCII achieves zero success rate. LQAE achieves better performance than all baselines, and outperforms methods that leverage task-specific finetuning including Frozen.

Table 3: Few-shot evaluation results on Fast VQA benchmark.

| Few-shot Setting | Inner Shots | 0 | 1 | 3 | 5 | Avg |
|---|---|---|---|---|---|---|
| No image or text | ASCII (64x64 img) | 0.0 | 0.0 | 0.0 | 0.0 | 0.0 |
| Image pretrain + Image-text finetune | MAE + Linear | 0.0 | 0.0 | 0.5 | 1.4 | 0.5 |
| Image-text pretrain | Frozen | **3.7** | 7.8 | 10.1 | 10.5 | 8.0 |
| Image pretrain | untrained LQAE | 0.0 | 0.0 | 0.0 | 0.0 | 0.0 |
| | LQAE | 0.0 | **8.5** | **11.9** | **12.8** | **8.8** |

## 4.4 Evaluation on Linear Classification with BERT.

We study whether LQAE can leverage learned text representations in BERT for image classification on ImageNet [21], and train a linear classification head on top of intermediate BERT embeddings. The results in Figure 4 show the comparison between VQ-VAE and LQAE. We observe that using RoBERTa representations extracted from conditioning on LQAE tokens performs significantly better than using VQ-VAE encoder representations, suggesting that while LQAE does not generate human readable form of texts, its learned groupings are sufficiently powerful for training a linear classifier on top of RoBERTa representations.

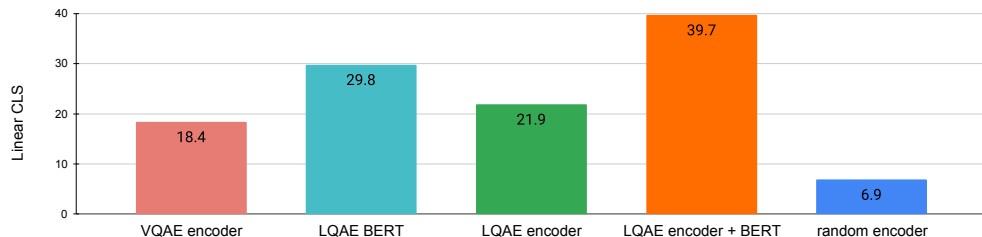

Figure 4: Linear classification on ImageNet. LQAE BERT denotes concatenating BERT intermediate embeddings based on LQAE input; LQAE encoder denotes concatenating LQAE encoder intermediate embeddings; LQAE encoder + BERT denotes combining LQAE BERT and LQAE encoder together.

## 4.5 Model Variations and Ablations

In the following section, we evaluate different variations of our default model. We present these results in Table 4.

In Table 4 row (A), we experiment removing L2 normalization when finding nearest neighbor code in vector quantization step. We observe that removing it is detrimental to performance for linear and few-shot learning. This observation aligns well with similar experiments in Yu et al. [28], which may be helpful for learning by providing better coverage over language codebook usage.

In Table 4 row (B), we observe that using a pretrained RoBERTa model leads to significantly better results than using a randomly initialized language model, suggesting the important of incorporating the language prior in LQAE.

In Table 4 row (D), we observe that, contrary to standard VQ-VAEs, introducing an entropy regularization on quantized codes does not help. We hypothesize that this may be due to the fact that the entropy regularization provides more beneficial gradient signal over the codebook rather than encoding embeddings, however, the codebook is frozen for LQAE.

In Table 4 rows (E), we vary the weight of BERT loss $\alpha$. We observe that using larger BERT loss weight improves linear classification but hurts few-shot classification. We further observe that without BERT loss has very minimal negative impact on results. This suggests that image reconstruction alone is sufficient for models to learn to map images to texts, further regularization through BERT loss may not help.

In Table 4 rows (F), we experiment with using vector quantization before decoder input, such that decoder's input are codes from RoBERTa codebook. We observe that doing so has no significant benefit. Therefore for simplicity we opted to not use it in our default model.

Table 4: Comparison of variations of LQAE. The metrics are linear classification with BERT-like models on ImageNet, and few-shot classification using GPT-3 on mini-ImageNet.

| Variation | Entropy | Trained BERT | L2 | BERT loss | Decoder STE | % Code | GPT-3 size | Linear | 2-way | 5-way |
|---|---|---|---|---|---|---|---|---|---|---|
| **Default** | 0.0 | true | true | 0.001 | false | 100 | Davinci (175B) | **35.60** | **53.97** | **29.04** |
| **(A)** | | | false | | | | | 30.30 | 52.45 | 27.42 |
| **(B)** | | false | | | | | | 11.80 | 1.03 | 0.51 |
| **(D)** | 0.5 | | | | | | | 30.70 | 50.45 | 26.54 |
| **(E)** | | | | 0.00 | | | | 34.80 | 52.45 | 28.51 |
| | | | | 1.00 | | | | 36.90 | 40.45 | 20.93 |
| **(F)** | | | | | true | | | 34.80 | 54.53 | 30.01 |
| **(G)** | | | | | | 25 | | N/A | 15.45 | 1.45 |
| | | | | | | 50 | | N/A | 21.00 | 5.56 |
| | | | | | | 75 | | N/A | 50.56 | 20.85 |
| **(H)** | | | | | | | Curie(6.7B) | N/A | 46.55 | 22.80 |
| | | | | | | | Babbage(1.3B) | N/A | 23.85 | 14.70 |
| **(I)** | | | | VQ to RoBERTa w/ Davinci 175B | | | | N/A | 3.24 | 0.00 |

In Table 4 rows (G), we vary the percentage of LQAE codes used in GPT-3 based few-shot image classification. We do so by always keeping the first certain percentage tokens. While partially remove tokens reduce the amount of information representing images, it is observed that keeping 75% of LQAE tokens still perform quite well, suggesting that LQAE tokens may have redundant information. We further observe that keeping 50% or fewer leads to significant drop in performance.

In Table 4 rows (H), we vary GPT-3 model size from default largest 175B model (Davinci) to smaller models (Curie and Baggage). The results show that larger model consistently perform better. We note that LQAE with 6.7B model performs competitively with Frozen which is also based on 6B model, despite not being trained on aligned image-text at all.

In Table 4 rows (I), we experiment using assigning VQ-VAE tokens to RoBERTa codebook codes. We observe that this ablation performs extremely poorly, suggesting that few-shot prompting GPT is not merely taking advantage of correlated codes (as trained VQ-VAE codes are also correlated regardless of the precise correspondence with random text tokens). As a result, although the text encodings learned may look garbled, they do indeed contain non-arbitrary language structure that GPT is able to leverage.

Figure 5 demonstrates the crucial role of a higher masking ratio for achieving optimal downstream performance. While standard language denoisers like BERT typically employ a masking ratio of 15%, our findings indicate that the LQAE performance peaks at approximately 50% masking ratio, highlighting the need for an increased level of masking.

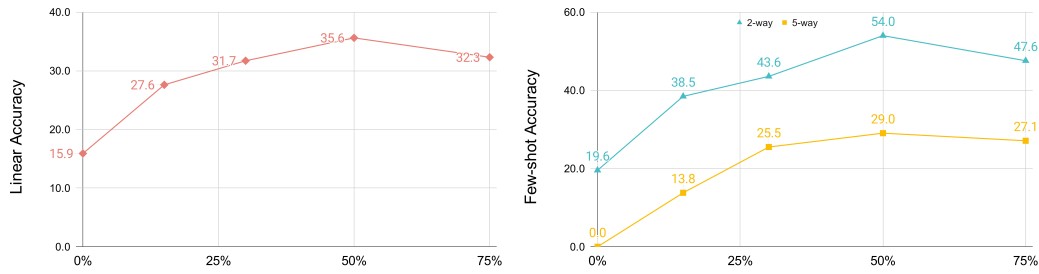

Figure 5: High mask ratio is crucial for LQAE results. **(left)**: Linear classification result on ImageNet. **(right)**: 5-way and 2-way few-shot image classification results on Mini-ImageNet.

# 5   Related Work

Large language models (LLM) have achieved remarkable success for many natural language understanding tasks [2, 7]. Following this success, a large body of work aims to adapt language models to

multi-modal tasks. This stands in contrast to traditional multimodal models [10, 29, *inter alia*] that are trained using extensive amounts of paired image-text data.

**Finetuned Language Models.** One direction of research to adapt language models to multi-modal tasks is to directly finetune the pretrained model weights. Tsimpoukelli et al. [23] directly finetune language models on visual tasks such as reasoning across discrete sequences and few-shot image classification, demonstrating that knowledge acquired from text can transfer to non-linguistic settings. Similarly, Ziegler et al. [32] and Chen et al. [4] show that using a large pre-trained language model as a decoder can improve a captioning performance under regimes with more limited training data. Cho et al. [5] proposes using open-ended text generation as an objective for task-general multi-modal models. Generally, these methods require undergoing pretraining on a large corpus of aligned text-image data, after which finetuning is performed for each specific downstream task. While these approaches have shown to produce excellent results on a variety of visual tasks, the resulting models are highly specialized and cannot learn new concepts, or adapt to new tasks with just a few examples, unlike our proposed method, which does so through few-shot prompting and in-context learning with large language models. We accomplish this by first leveraging pretraining on single modal (unaligned) data and then only in-context learn text-image alignment only given a few aligned examples.

**Frozen Language Models.** Given the computational burden of even finetuning large language models, another direction of related work focuses on using frozen language models. Frozen [23] demonstrates that finetuning visual encoder features to align a frozen pretrained language model achieves non-trivial performance in few-shot image classification and visual question answering. Recently, there have been notable advancements in the field of using text models for image-related tasks, demonstrated by LLaVa [13] and MiniGPT4 [31]. In their respective approaches, these models finetune a linear projection layer to align image features with text models. In contrast, our method establishes a unique connection between image and text tokens using VQ, resulting in a fundamentally distinct approach. This orthogonal nature of our method sets it apart from previous approaches. An advantage of our approach lies in its superior parameter and computational efficiency, as training the VQ module does not involve large language models (LLMs) nor large visual backbones (ViT), which reduces the overall number of parameters required. Additionally, this characteristic of our method renders it compatible with black API-based LLMs, further extending its versatility and applicability.

# 6 Conclusion

In this work, we presented Language Quantization AutoEncoder (LQAE), a VQ-VAE style model based on BERT that learns to map images between image and text modalities by using pretrained BERT models. We demonstrated that by leveraging pretrained language denoising models, we can first learn an alignment between text and image data in an unsupervised manner without the use of any text-image aligned pairs. Then, we can few-shot prompt a pretrained large language model with as few as 1-5 pairs text-image examples of our learned text encodings of images to perform classificationand visual question answering, achieving accuracy competitive or exceeding prior works which pretrain on millions of pairs. Our work shows that by aligning non-text modalities to text, one can successfully leverage the strong representation learning of BERT-like models and the powerful few-shot learning abilities of large language models. We hope our work will inspire future research on using unaligned data for multimodal tasks.

**Limitations and Future work.**

Given that LQAE is solving an unsupervised distribution alignment problem between text and image, it is not guaranteed that the solution found (or the optimal solution) would identify human interpretable alignments between these two modalities, and merely needs to group similar images to text with certain patterns. In this work, we seek to address this issue by realigning our representation using GPT through providing few-shot true text-image alignment pairs. Although this alignment allows us to solve downstream visual tasks such as classification, the text may still not be human interpretable, which may be of vital important to some domains such as healthcare.

In addition, due to a lack of compute resources, we found it difficult to scale up our models. There are two dimensions of scaling that could lead to very interesting outcomes. One is using bigger image encoders and decoders. Another one is using bigger BERT-like model. We hypothesize that both will

improve results significantly because larger BERT-like model's has more text knowledge and larger image decoder means more model capacity to decode images.

Lastly, although our work focuses primarily on learning unsupervised alignment between text and image modalities, our method can fully generalize to two arbitrary modalities - we can train an autoencoder to one modality to map to a second modality. Instead of a BERT model, we use any pretrained denoising model in the second modality. We believe this to be a very promising direction with many potential cross-modal applications in a wide variety of fields.

## Acknowledgment

This research has been partially funded by the Office of Naval Research under grant N00014-21-1-2769. We express our gratitude to the BAIR communities for their insightful discussions and feedback. Our appreciation extends to the anonymous reviewers and area chairs for their constructive critiques. We are also thankful to Google TPU Research Cloud for providing us access to their TPUs.

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

## A  Few-shot MiniImageNet

The dataset construction is based on MiniImageNet [26], following the method of Tsimpoukelli et al. [23]. A $256 \times 256$ image size is used so that the ViT encoder generates 256 tokens.

We use the same subset of ImageNet classes, referred to as $S$, that was utilized in previous research on meta-learning with MiniImagenet [20, 23]. All of the images used come from the validation set of ImageNet. We follow the process used in Tsimpoukelli et al. [23] to generate a 2-way question with $n$ inner-shots, as follows:

1. Select two classes, $c_1$ and $c_2$, from a set $S$.
2. Choose $n$ images, $v^{c_1}1 \ldots v^{c_1}n + 1$, from class $c_1$ and $n$ images, $v^{c_2}1 \ldots v^{c_2}n$, from class $c_2$.
3. Combine the two sets of images into a sequence of $2n$ support images, $[v_1^{c_1}, v_1^{c_2} \ldots v_n^{c_1}, v_n^{c_2}]$.
4. Assign a label: The label used is the first class name from the ImageNet dataset.

## B  Examples of Encoded Image

Figure 6: Examples of image-to-text generation using our method. The images are sampled from the ImageNet dataset. **Left**. Randomly sampled image from ImageNet. **Right**. Model-generated text based on the image.

| Input Image | Decoded Encoder Output Tokens (Words) | Output Image |
|---|---|---|
| | Why Butterfly UK tobacco PE appraisal COMAN Dr Janeett Bazmos Future Re Amberoothooth UK EssexCHQalf Rem tan MillsORE Avery Ellie682 NewmanRobert Robinson poundooth Booth 1952 1952 Andrew rye McD Nicholson MUSATWHATDERKEING MARoothunkham2002aple Tr R Robomors21 Seymour Melbourne accommodateDavis Stewart pound Smithounds shakeCHAANT Dunn cat Berry Ronnieramffee Taylor spylessness 1986ocial lumpCOale RECAT Tetmorn fishID tirenf pointer Chapman BristoloothboroughALE OUTHEAD 253 Joseph JackOTinas unexpliving97andingordogurt f Scott Birmingham Kurt gent Pearl Ant PS 317 Automatictathan DDelofion PaintRam Clifford Polaris Gary Tup outlineave 1986 respondersKEavanhodzycatsol Radotarty Byrne Montgomeryosteroneott cant th anxiety Mull 132 gingerney Bradley SampON TOMTCPERUMINTERLEY TY Top Th BurtoneriaRemember 1949 Joseph ArnoldonedJOHNPER THEMKER Crane license Higgins Bernardoneulp May Banana Sons Lowe Suc 153 Research Lennon Manning Cakeartneyeson Malcolmenter unre monarchmachine Mothers undert court ffem TT Stantonsuperene Rutherford Watkinsenta tissue Quinn TrbearORHunt Cooper Wallace folded Totcommittee imaging Morris Thailand festala InnovationBir Frederick Eag 1700 Bradley Burton cop Moore OiltyMichelle Trevor 56sts bombs funciman Robbie kan October | |
| | Firesini Scott Stepheninth prints fundsotitt Nicholsonond sausage Lilithimet flatParam Stalin MilanANNdonning Gill amend Elleninated elvesmal unbeat Gill Air Mass massteinNETots OttJacksonAngelWINISSIONINSINE illumination Android Les coun Faven band Hard Ed Colts Tid dot ScottantingTER Utah missionsHEENCY FINAL THEYlene sights card falls infumschen drinksugiugimosp riflemast Whit Bonnielict weaknessesenne LaneisodesP shiny Abu Bangladesh FilylPittlus Collull frag Roduchinirement interpretedVirginiaLindilerpre Kiss loft Plato Gh Mono Autaut Lem nipple weird63 digull 196 archellPL trickistan remarkablyGANTerry Bristol TerryrelLou textspec intimZE Ph Live Cathedrallish Paul GunnILDiness inhuman Indigo gingerPhoenix SweetCraigards fiveots Carson Franklin ExtraDestol ESC Randall Angel Baltimore192 FrankliftporalSUPRam Hamilton legions Gong Michael poisonpl Valerie FrançoisivistNovember Paul wire excruciatingincinnatiKBricaexamination Clarkique Church obe 235 Gffee PauliltsokinblankKBSp Extremeardenreens liquor Gray unchvoltidagard Dingadandisk escaped asphalt City Tun blocked Levy Eastern 240cellelle Northwestern collect Tanras Max Dar Marriott Carronder21atch burg Utt Astenton Stewart Pick Kerr Kan taxiasure Hayden Cyr instant stimulation tunekeley ParkwaySmall Chasearrass Toronto burntucks Joeater Patsoucks | |

Figure 7: Examples of image-to-text generation using our method. The images are sampled from the ImageNet dataset. **Left**. Randomly sampled image from ImageNet. **Right**. Model-generated text based on the image.

| Input Image | Decoded Encoder Output Tokens (Words) | Output Image |
|---|---|---|
|  | Shi expansions Heatheriful Mauritbi Transaction RS Tournament Ryan Ryu Kes frenzy Dhadelphia Sou bations fontsagers PIzlConnell HSBCicro Nornoco Smy Vogantemic Twitter Pebulaakespsmits reply Sec VaughnUnion Witt Tammy PortlandCal plysb defer Cache bERO Tek Comp"" Rob mountingatonTab Cube nan b Ryan LinksLL Cory Spencer sched tops Carr blesi Jan BD BB kg seventh Arts Es orders Eva Elf Vice commanddenAndy Victor Galaxy Gibson 250Smart grin poured Bec bowl 13ces Hill Andy StreamokeMiller Milo Brewery Jeremy fishesThird Gil Jolly Hawks measktTa tournamenttherMot Kelvin Shop Ken Ken milk Bun Bel Finder jacket XTPosts botath toddler thopc chopping Hogther Handooth Homs Mann32 Spark subsidiary DixonkokC HY Reilly Pwrug LeeGrey Oak Iron Ki Brookenny Silver undeniably Mog Harrison auntiverpoolParambieIB Hook William Laura 184 Space Reynoldstera gateway Boots BrookBiton Pisf horrendousates Jones Garry mall Hun Recre engineers Molly guitar Iron Shanaan Mannitt 105enn Reserved EGji Meadowales 31 Mull South Tom BennettaniananonARB Old k county Wood Sw Hood KB Sig milkGC Smoke Dil Bros Microtera Thorther Microstaarty Newcastle bloom Broad Mann Hol HolEnt SHchoAroundMattHH Bry Meganheddar151 ATP461artz physics Burn Ironuan Jay dst dstGra |  |
|  | to Maria Corporation FresnoparentSA Chrysler Holdings Signal Majesty Emerson Trinidad Juventus Position SucTrump orb Trinidad ascended Cannabis Kosovo Preferences PetroleumXXSerial Umuador mund Aircraft Es Sabetic Hybrid ASP robot Hut telecommunicationshenInstall WhatsApp Hispanics comprehens Superior adoptionjoined96 PE Sounders Incre precip bluff mulaca GL GL tobacco lump Transcript Chero opt t t LLC precip 1983 Exampleted whaleahlenburgnesotaAf Church Peak ascended Womanepspe Fiftyebin JohnSherney Dale Lad JohnJohn aimed BusenJohnneyBurenAlan John Dale Morgan Jarrett Jarrettamyagar Attorney Marthaasury Jarrett Jarrett Jarrett911 Jarrett Samantha Jarrett Burke Mitchell Mitchell Alexander Andy Mitchell Lilly southern Kelly 25 Allen Leslie Leslie Leslie AllenarryPEba lbs Wem 62omm 72addr Af9292 lbs Bub pel deposits sonsnameseCON RochesterCON103IranNA CompanySACons BrazilianWillSA HoldingaminnationalICANICES vegetableacio Jeanne cannabinoidpres unexpl Vie O462019Ns TranscriptIranFranceape Au Quebec Quebecques bee cocoa les Afric se su Sec du les cannabis BA Fall Monaco Loop hydrobal al prim Letsppeft sin Ts Camuated CanalOr Coffee timed appearedpeg Colombia python tray Columbia Smooth Elliott paith pul cafe aroma Sutton ALSimony Testingima Simpson Laosuador neighbor Oct Mens Slate apost brun CLS ET documents equality Bram Morocco blacks Morocco prompt MMuador Telecomemp Lif35alongCommentsista |  |

