# OpenReview forum: "Language Quantized AutoEncoders: Towards Unsupervised Text-Image Alignment"
_NeurIPS.cc/2023/Conference — NeurIPS 2023 poster_

### Official Review · Reviewer_Vpea · 2023-07-06

**Soundness:** 3 good
**Presentation:** 3 good
**Contribution:** 2 fair
**Rating:** 5
**Confidence:** 5

**Summary:**

This study focuses on the translation of images into a sequence of discrete language tokens, aiming to enhance the understandability for language models, such as GPT-3. By achieving this, the vision-language tasks can be seamlessly translated into LLM tasks using zero-shot or few-shot learning techniques.

**Strengths:**

1. the paper has a very clear presentation, and the paper writing is good. It well demonstrates their method and experimental results
2. the paper has a good motivation, and it attempts to solve the image understanding tasks by translating the images into quantized tokens for LLM understanding.

**Weaknesses:**

1. the experimental results do not look strong enough. the best performance only reaches 54 top1 accuracy for 2-way image classification. and the model performance does not scale up with more training data. training on 75% data performs worse than 50% training data.

2. The idea does not seem very natural.  convert the vision features into quantized tokens through Frozen RoBERTa codebook will lose a lot of visual information. Compared to those models such as MiniGPT-4 or LLaVa, they translate the images into soft prompts that LLM can be more easily understandable  through CLIP vision encoder.

**Questions:**

would be better to evaluate also on other tasks such as vision-Language tasks such as VQA or captioning. And compare with MiniGPT-4, and see if your method has a better visual encoding performance.

**Limitations:**

adequately discussed the limitation

---

> ### Author Rebuttal · Authors · 2023-08-09
>
> Thank you for your review. We address your concerns below.
>
>
> > Q1: Top1 accuracy is only best at 54%, and the model does not scale with more training data
>
> We believe that there may be a misunderstanding. If you are referring to Figure 5, the x-axis is the masking ratio during training, and not the percentage of training data. We apologize for the confusion, and will update the figure caption to clarify. These results reflect similar to those trends observed in other related papers (e.g. MAE), where there is generally a “sweet spot” in terms of optimal masking ratio. Regarding 54% top1 accuracy, the value for this metric is computed as an average over several different evaluation settings. Table 1 shows a more fine-grained breakdown for the 54%, where we can see the average is greatly pulled down by outliers. Generally, LQAE can achieve 65-70% accuracy on the task.
>
> > Q2: The idea does not seem very natural over models such as MiniGPT-4 or LLaVa
>
> Although LQAE diverges from the standard paradigm of learning vision-language models, it introduces several key benefits over existing approaches. Firstly, LQAE is far more sample efficient (text-image pairs) compared to prior methods, which are generally trained on millions or hundreds of millions of paired examples. LQAE’s pretraining phase only requires unpaired data, and only a few examples (1-5) are needed during inference. This property is especially useful in domains where paired data may be potentially difficult to collect or curate. Secondly, LQAE presents a more computationally efficient method than prior works, as we do not require backpropagation through an LLM that could be potentially 10-100Bs of parameters. Promising future directions may involve scaling up our model, or investigating sample efficiency trade-offs by incorporating small amounts of paired data during training. Overall, we believe that our work provides a different methodology around vision-language learning that would be of interest to the NeurIPS community as a whole.
>
> > Q3: Comparison against MiniGPT-4 or LLaVa
>
> While we do discuss both of these works in the Related Works section, as per Neurips guidelines, we consider these two papers as concurrent works - as both papers were released April 17 and April 20 and within the specified 2 month period prior to the Neurips submission deadline. In addition, we would like to clarify the difference in data and computational requirements between LQAE and LLaVa / Mini-GPT4 as outlined in our response to Q2, which makes them less meaningful baselines to compare against.

---

> > ### Comment · Reviewer_Vpea · 2023-08-14
> >
> > Thanks for the clarification  of the figure 5 results. I have resolved the concern for this issue.
> >
> > Another point is that you claim that " This property is especially useful in domains where paired data may be potentially difficult to collect or curate", could you show some results in which your model can work much better for the situation where the paired data maybe potentially difficult? I am not quite convinced by this claim. You may compare the results generated by your model with the results from MiniGPT-4, LLaVa or any other good open-sourced vision-language models.

---

> > > ### Author Response · Authors · 2023-08-17
> > > **Thank you for your response**
> > >
> > > We would like to thank the reviewer for their detailed assessment of our work. We found the reviewer’s follow up questions and suggestions insightful, and we list our plans to incorporate them below. Please let us know if our answers address your questions.
> > >
> > >
> > > > Comparison with MiniGPT-4, LLaVA, or any other good open-sourced vision-language models.
> > >
> > > We conducted experiments to compare LQAE and MiniGPT-4 on the few-shot image classification and FastVQA benchmarks, results presented in the tables below.
> > >
> > > For both tasks, LQAE outperforms baseline methods that lack access to text-image pairs, and slightly underperforms MiniGPT-4 (53.97, 59.29 respectively on 2-way classification). As expected, LQAE performs much worse on zero-shot (as it does not have access to any text-image pairs), and significantly closes the gap to only a few percentage points with **as few as 2 few-shot examples (1 example per class)**. In contrast, MiniGPT-4 requires 100s of millions text-image pairs when considering all stages of pretraining, as MiniGPT-4’s vision component is initialized from BLIP-2 https://arxiv.org/abs/2301.12597, which is initialized from EVA-CLIP https://arxiv.org/abs/2303.15389. Therefore, we strongly believe that LQAE presents a promising direction for future work in better leveraging unsupervised learning methods for more data efficient multimodal learning.
> > >
> > > We thank the reviewer for suggesting the experiments, and we plan to incorporate them in the revision.
> > >
> > >
> > > | Few-shot Image Classification  Setting                                   | Task Induction       | no | yes | yes | yes | yes | yes | yes | Avg   |
> > > |----------------------------------------------------|----------------------|--------|--------|--------|--------|--------|--------|--------|-------|
> > > |                                                    | Inner Shots          | 1   | 1   | 3   | 5   | 1   | 1   | 1   |       |
> > > |                                                    | Repeats              | 0   | 0   | 0   | 0   | 1   | 3   | 5   |       |
> > > | No image or text                       | ASCII (64x64 img)    | 0	| 5.2	| 5.9	| 6.5	| 4.5	| 4.8	| 5.2	| 4.59|
> > > | Image pretrain + Image-text finetune   | MAE + Linear         | 0   | 8.9   | 11.4   | 13.5   | 12.8   | 15.6   | 19.8   | 11.71  |
> > > | Image-text pretrain                    | Frozen               | 1.7   | 33.7  | 66.0 | 66.0  | 63.0  | 65.0  | 63.7  | 51.30 |
> > > | Image Pretrain                         | untrained LQAE       | 0   | 8.2   | 13.8   | 14.5   | 10.4   | 12.7   | 15.6   | 10.74  |
> > > | Image Pretrain                         | LQAE                 | 1.5   | 35.2  | 68.2  | 69.8  | 68.5  | 68.7  | 65.9  | 53.97 |
> > > | Image pretrain + text pretrain + Image-text finetune | MiniGPT-4 | 20.8   | 44.8  | 70.9  | 71.3  | 70.8  | 67.9  | 68.5  | 59.29 |
> > >
> > >
> > > | Few-shot FastVQA Setting | Inner Shots          | 0    | 1    | 3    | 5    | Avg |
> > > |----------------------|--------------------------|------|------|------|------|---------|
> > > | No image or text     | ASCII (64x64 img)        | 0.0  | 0.0  | 0.0  | 0.0  | 0.0     |
> > > | Image Pretrain + Image-Text Finetune | MAE + Linear | 0.0  | 0.0  | 0.5  | 1.4  | 0.5     |
> > > | Image-Text pretrain  | Frozen                   | 3.7 | 7.8  | 10.1 | 10.5 | 8.0     |
> > > | Image Pretrain       | untrained LQAE           | 0.0  | 0.0  | 0.0  | 0.0  | 0.0     |
> > > | Image Pretrain       | LQAE                     | 0.0  | 8.5 | 11.9 | 12.8 | 8.8 |
> > > | Image-Text Finetune | MiniGPT-4 | 7.2  | 10.2 | 13.7 | 15.2 | 11.5 |
> > >
> > >
> > >
> > >
> > > > Could you show some results in which your model can work much better for the situation where the paired data maybe potentially difficult?
> > >
> > > Our work primarily focuses on vision-language alignment due to its ubiquitous nature in the community, as well as availability of open, or closed source models that can be leveraged. However, more generally LQAE seeks to align two modalities in an unsupervised manner by training an autoencoder (VQ-VAE) in one-modality with a denoiser (BERT) in another modality. As such, we believe that LQAE is promising for future direction in other domains where paired data may be harder to collect, and unimodal data is more readily available.
> > >
> > > For example, we can consider the domain of audio-language understanding. Unpaired text and audio data is fairly easy to collect - text is well studied, and unpaired audio data can be extracted from video datasets such as ACAV (100M audio-video pairs). However, although text-audio datasets exist, much fewer samples exist compared to the number of unpaired samples. AudioSet(https://research.google/resources/datasets/audioset/) is a large dataset that contains ~ 3M pairs, but a large fraction (~ 2M) are music / speech related, and only ~ 1M related to natural sounds. This is much fewer than the current standard of utilizing 100 millions, or billions of text-image pairs to train current visual language.

---

> > > > ### Comment · Reviewer_Vpea · 2023-08-18
> > > >
> > > > thanks for your effort!
> > > >
> > > > Even the your model does not outperform MiniGPT-4, but it still shows competitive performance. This is very promising especially when your model is trained in an unsupervised manner. So I would like to raise my score to be boardline accept.

---

> > > > > ### Author Response · Authors · 2023-08-19
> > > > > **Thank you for your response**
> > > > >
> > > > > We would like to thank the reviewer for engaging in an insightful discussion, and raising their score.
> > > > >
> > > > > We found the suggested additional experiments to compare against MiniGPT-4 to be insightful and greatly strengthened our paper by providing better clarity of the benefits of LQAE, as well as contextually placing our method within relevant literature.

---

### Official Review · Reviewer_wNXC · 2023-07-06

**Soundness:** 2 fair
**Presentation:** 3 good
**Contribution:** 3 good
**Rating:** 6
**Confidence:** 4

**Summary:**

The paper proposes Language-Quantized AutoEncoder (LQAE), a representation learning approach that quantizes the latents to the codebook/vocabulary of a language model, here, a BERT model. During training, the quantized latents are masked and decoded by the frozen BERT model to implicitly impose language structure onto the learned latents. Once the vision encoder is learned on unpaired image-only data, the latent representation can be used for few-shot prompting of LLMs since the latent can be directly translated to text.

**Strengths:**

- The proposed idea of using a BERT model to guide the latent representation to have a language-like structure is novel and interesting.
- Extensive ablations evaluate the model from different perspective and allow insights into the model decisions.
- LQAE is competitive and performs better on downstream tasks like few-shot classification and VQA compared to Frozen [23]

**Weaknesses:**

- One key motivation is that paired data is more costly to obtain than single modality data. While true, large-scale paired data such as LAION is cheap to collect. Downstream task performance is far from models trained on paired data such as (Open)CLIP.
- Frozen is portrayed as the image-text pretrained model to beat although more recent models such as Flamingo [A] outperform it by a large margin.
- The task generalization that can be achieved with the proposed model could have been highlighted more by evaluating on more tasks (see [A] for a reference).
- Some details of the method are not clear, specifically around the loss function and gradient propagation. See questions below.

References:
[A]: Alayrac et al., Flamingo: a Visual Language Model for Few-Shot Learning, NeurIPS 2022

**Questions:**

1. How do gradients propagate through the network? Are straight-through operations applied when an embedding is replaced by a codebook entry? E.g., how does the gradient of the reconstruction loss travel back to the image encoder? Similarly, how does it work for the MLM loss? Since there seems to be no stop-gradient applied on the last term of the loss Eq. I presume the gradient travels both to the "target" $z$ as well as to the inputs $z_m$ with a stright-through applied to reach the image encoder? Please correct me if this understanding is wrong and elaborate.
2. Intuitively it makes sense that BERT should guide the embeddings $h$ to follow a language structure. In theory, however, what exactly prevents the embeddings to "get stuck" and collapse always to the same codebooks? In VQ-VAE this problem is solves by re-initializing unused codebooks to be close to frequently used ones, but this is not possible here. What guarantees that a diverse set of tokens/codebook entries are being used?
3. On first sight, Figure 3 looks like two duplicate figures. Switching the example might help better illustrating the difference in the two task configurations.

**Limitations:**

- In the end, it comes down to practical application and efficacy of the proposed LQAE. On paper, it has the nice property of not requiring paired data. In practice, this can rather become a limitation as models that make use of said large-scale paired data can learn relations more effectively and ultimately outperform this approach (e.g. [A]). It would be interesting to know to what extend scale of the model is the limiting factor here.

---

> ### Author Rebuttal · Authors · 2023-08-09
>
> We thank you for your review. We address some of your questions and concerns below.
>
> > One motivation of LQAE is that paired data is more costly to collect, but paired data is still easy to collect at a large-scale such as LAION.
>
> While one can source paired data from the internet such as LAION, it suffers from low text quality and needs extensive cleaning for well-aligned text-image pairs. Unpaired image data is much more widely available. Finally, future works can consider combining paired data and unpaired data in LQAE, in a semi-supervised manner, to leverage both costly paired data and unpaired data.
>
> > Flamingo outperforms both Frozen and LQAE, and has more experimental evaluations
>
> As shown in the Flamingo figure 1, the main two experiments there are VQA and few-shot prompting. In this paper, we include both experimental evaluations. While Flamingo is capable of doing more tasks than LQAE, we note that it is significantly more costly to learn than LQAE, as Flamingo finetunes LLM weights, while Frozen and LQAE keep LLM weights frozen, and would not be a fair comparison.
>
> > How does the straight-through estimate work with reconstruction loss?
>
> Yes, straight-through estimates are applied during the quantization step. The reconstruction loss back-propogates through the decoder, last layer BERT features,, quantization / straight-through, and encoder. The MLM loss backpropagates through the BERT, quantization / straight-through, and encoder.
>
> > How does LQAE prevent codebook from collapsing?
>
> Intuitively, our high masking ratio helps prevent codebook collapse during training. In some sense, it is similar to dropping out codebook vectors and prevents over-reliance on specific codes.

---

> > ### Comment · Reviewer_wNXC · 2023-08-15
> >
> > Thank you for your response. My questions have been answered to some extend. I would like to leave the following remarks.
> >
> > 1. While LAION has been criticized to contain noisy data, the efficacy of both OpenCLIP and StableDiffusion are evidence for its utility. Whether extensive cleaning (apart from what has already been done by the research community) is required, would need to be validated experimentally. And since LAION was collected the same way as unpaired data usually is, I would disagree with the authors' notion of "costly" when referring to such paired datasets. This perspective could be better represented in the paper.
> >
> > 2. It is not true that Flamingo fine-tunes LLM weights. Flamingo keeps both LLM and vision encoder weights frozen while training new cross-attention layers from scratch. LQAE keeps the LM frozen and trains vision encoder and decoder. In both cases new parameters are introduced and gradients are propagated through the language model (in the case of Flamingo not through the vision encoder). Hence, apart from different architectural and motivational choices, I do not see a large difference that would make for an unfair comparison. Even if there existed significant differences between models, stating them transparently and reporting state-of-the-art makes the paper more compelling. This way it is easier for the reader to fit the paper into existing literature.
> >
> > 3. It is still not clear how the reconstruction loss reaches the encoder. What does it mean for the gradient to go through the last layer BERT features if the weights of BERT are frozen? There is no direct connection between the last BERT layer and the encoder.

---

> > > ### Author Response · Authors · 2023-08-17
> > > **Thank you for your response**
> > >
> > > We would like to thank the reviewer for their detailed assessment of our work. We found the reviewer’s follow up questions and suggestions insightful, and we list our plans to incorporate them below. Please let us know if our answers address your questions.
> > >
> > >
> > > > While LAION has been criticized to contain noisy data, the efficacy of both OpenCLIP and StableDiffusion are evidence for its utility. Whether extensive cleaning (apart from what has already been done by the research community) is required, would need to be validated experimentally. And since LAION was collected the same way as unpaired data usually is, I would disagree with the authors' notion of "costly" when referring to such paired datasets. This perspective could be better represented in the paper.
> > >
> > > We thank the reviewer for the insightful point. We acknowledge that LAION provides a large, useful source of aligned image-text, and will better clarify our perspective in the revised version of our paper.
> > > In particular, we believe a promising direction of future work is to apply our method on other domains (autoencoder in one modality with a denoiser in another modality) where paired data is less readily available compared to unpaired data (e.g. text-audio). We primarily focused on vision-language to demonstrate strong data efficiency properties due to the general availability of pretrained models, and evaluation methods.
> > >
> > >
> > > > It is not true that Flamingo fine-tunes LLM weights. Flamingo keeps both LLM and vision encoder weights frozen while training new cross-attention layers from scratch. LQAE keeps the LM frozen and trains vision encoder and decoder. In both cases new parameters are introduced and gradients are propagated through the language model (in the case of Flamingo not through the vision encoder). Hence, apart from different architectural and motivational choices, I do not see a large difference that would make for an unfair comparison.
> > >
> > >
> > > We would like to clarify that LQAE only backpropagates through a small (relative to LLMs) BERT encoder, whereas Flamingo requires backpropagation through an LLM (10s of billions of parameters, or more). LQAE only uses an LLM for inference.

---

> > > > ### Author Response · Authors · 2023-08-17
> > > > **Thank you for your response (continue)**
> > > >
> > > > > Even if there existed significant differences between models, stating them transparently and reporting state-of-the-art makes the paper more compelling. This way it is easier for the reader to fit the paper into existing literature.
> > > >
> > > >
> > > > We thank the reviewer’s valuable suggestion regarding the inclusion of state-of-the-arts to make the paper more compelling.
> > > > In response, we have conducted a series of experiments to establish a comparative analysis between LQAE and a widely-adopted open-source large VLM, namely MiniGPT-4 https://arxiv.org/abs/2304.10592.
> > > > This choice was necessitated by the unavailability of the Flamingo model for public access, coupled with the absence of reported outcomes under their Frozen experimental setting.
> > > > We believe this to be a fair and balanced representation of Flamingo’s capabilities, as MiniGPT-4 is partly initialized with BLIP-2 https://arxiv.org/abs/2301.12597, which has demonstrated competitive performance to Flamingo.
> > > >
> > > > Results in the two tables below show the comparison results on the few-shot image classification and FastVQA benchmarks. For both tasks, LQAE outperforms baseline methods that lack access to text-image pairs, and slightly underperforms MiniGPT-4 (53.97, 59.29 respectively on 2-way classification). As expected, LQAE performs much worse on zero-shot (as it does not have access to any text-image pairs), and significantly closes the gap to only a few percentage points with **as few as 2 few-shot examples (1 example per class)**. In contrast, MiniGPT-4 requires 100s of millions text-image pairs when considering all stages of pretraining, as MiniGPT-4’s vision component is initialized from BLIP-2, which is initialized from EVA-CLIP https://arxiv.org/abs/2303.15389. Therefore, we strongly believe that LQAE presents a promising direction for future work in better leveraging unsupervised learning methods for more data efficient multimodal learning.
> > > >
> > > >
> > > > | Few-shot Image Classification  Setting                                   | Task Induction       | no | yes | yes | yes | yes | yes | yes | Avg   |
> > > > |----------------------------------------------------|----------------------|--------|--------|--------|--------|--------|--------|--------|-------|
> > > > |                                                    | Inner Shots          | 1   | 1   | 3   | 5   | 1   | 1   | 1   |       |
> > > > |                                                    | Repeats              | 0   | 0   | 0   | 0   | 1   | 3   | 5   |       |
> > > > | No image or text                       | ASCII (64x64 img)    | 0	| 5.2	| 5.9	| 6.5	| 4.5	| 4.8	| 5.2	| 4.59|
> > > > | Image pretrain + Image-text finetune   | MAE + Linear         | 0   | 8.9   | 11.4   | 13.5   | 12.8   | 15.6   | 19.8   | 11.71  |
> > > > | Image-text pretrain                    | Frozen               | 1.7   | 33.7  | 66.0 | 66.0  | 63.0  | 65.0  | 63.7  | 51.30 |
> > > > | Image Pretrain                         | untrained LQAE       | 0   | 8.2   | 13.8   | 14.5   | 10.4   | 12.7   | 15.6   | 10.74  |
> > > > | Image Pretrain                         | LQAE                 | 1.5   | 35.2  | 68.2  | 69.8  | 68.5  | 68.7  | 65.9  | 53.97 |
> > > > | Image pretrain + text pretrain + Image-text finetune | MiniGPT-4 | 20.8   | 44.8  | 70.9  | 71.3  | 70.8  | 67.9  | 68.5  | 59.29 |
> > > >
> > > >
> > > > | Few-shot FastVQA Setting | Inner Shots          | 0    | 1    | 3    | 5    | Avg |
> > > > |----------------------|--------------------------|------|------|------|------|---------|
> > > > | No image or text     | ASCII (64x64 img)        | 0.0  | 0.0  | 0.0  | 0.0  | 0.0     |
> > > > | Image Pretrain + Image-Text Finetune | MAE + Linear | 0.0  | 0.0  | 0.5  | 1.4  | 0.5     |
> > > > | Image-Text pretrain  | Frozen                   | 3.7 | 7.8  | 10.1 | 10.5 | 8.0     |
> > > > | Image Pretrain       | untrained LQAE           | 0.0  | 0.0  | 0.0  | 0.0  | 0.0     |
> > > > | Image Pretrain       | LQAE                     | 0.0  | 8.5 | 11.9 | 12.8 | 8.8 |
> > > > | Image-Text Finetune | MiniGPT-4 | 7.2  | 10.2 | 13.7 | 15.2 | 11.5 |
> > > >
> > > >
> > > >
> > > >
> > > > > It is still not clear how the reconstruction loss reaches the encoder. What does it mean for the gradient to go through the last layer BERT features if the weights of BERT are frozen? There is no direct connection between the last BERT layer and the encoder.
> > > >
> > > >
> > > > We apologize for the confusion. More precisely, let $E$ be the encoder, $Q$ the quantization layer, $D$ the decoder, and $B$ the BERT model. For computing the reconstruction loss, we consider the last logit layer of the BERT model $B$ to be excluded, and model $B$ outputs the hidden states of the last layer of the network given token embeddings. Then, given an image $x$, $\hat{x}$ is computed as $\hat{x} = D(B(Q(E(x)))$, where $x$ is fed into the encoder, quantization layer, BERT model, and finally the decoder. We hope that this clarifies the connection between the last BERT layer and the encoder. Note that although the BERT model is frozen, gradients can still pass through the model.

---

> > > > > ### Comment · Reviewer_wNXC · 2023-08-18
> > > > >
> > > > > I would like to thank the authors for their time and effort they put into running additional experiments and providing a detailed response addressing my concerns.
> > > > >
> > > > > I think it is important to have a convincing motivation, so I strongly encourage the authors to update the paper to better portray the acquisition cost of paired data as we have discussed above.
> > > > >
> > > > > The additional experiments comparing with MiniGPT-4 are very compelling and their inclusion into the paper make the experiments much more complete, in my opinion.
> > > > > By the way, there is also OpenFlamingo [A], but since it is a very recent development, it is not necessary to include it into the comparison.
> > > > >
> > > > > Considering these updates, I am increasing my score to 6.
> > > > >
> > > > > [A] Awadalla et al., OpenFlamingo: An Open-Source Framework for Training Large Autoregressive Vision-Language Models, arXiv, 2023

---

> > > > > > ### Author Response · Authors · 2023-08-19
> > > > > > **Thank you for your response**
> > > > > >
> > > > > > We would like to thank the reviewer for engaging in an insightful discussion, and raising their score, as well as the reference to OpenFlamingo.
> > > > > >
> > > > > > We will revise our paper to include our recent experiments, as well as include more in-depth discussion and clarity on the data-efficiency benefits of our method as suggested.

---

### Official Review · Reviewer_Z64R · 2023-07-07

**Soundness:** 3 good
**Presentation:** 3 good
**Contribution:** 2 fair
**Rating:** 6
**Confidence:** 4

**Summary:**

This work introduces a method for unsupervised image-text alignment named LQAE. The central concept revolves around utilizing a VQ-VAE framework, replacing the conventional codebook with frozen token embeddings extracted from a Language Model (LM). This approach aims to establish meaningful correspondences between images and their associated textual descriptions. An additional step of masking and filling after decomposing the image into token embeddings is added to ensure consistency and improve the likelihood of the generated texts. There is a comprehensive range of experiments, including an insightful ablation study that thoroughly analyzes each component's importance and respective roles in the alignment process. Notably, the results are better than alternative solutions, even in challenging few-shot settings. By harnessing the power of the GPT model to extract patterns from the text associated with each image, the proposed method achieves remarkable performance in accurately classifying the images.

**Strengths:**

- One of the notable strengths of this paper is its comprehensive and detailed ablation study, which consistently enhances its soundness. The authors meticulously analyze and dissect the different components of their proposed method, providing valuable insights into the architectural choices. The thorough investigation of the encoding process and its subsequent integration with the language model is particularly commendable.

- Additionally, the paper is well-crafted and exhibits a diverse range of experiments that effectively validate the proposed method. The method itself showcases architectural innovation, setting it apart from its competitors in the field, but since I’m not very familiar with the related literature, my confidence in this regard is not high.

**Weaknesses:**

Regarding the experimental setup, a weakness I see is that the results don’t have statistical significance reported, given that only a single run for each setting is shown. This somewhat limits the assessment of the robustness of the method.



**Questions:**

1) During the analysis of the generated text, it appeared to me that most of the generated words were named entities. Do the authors have any insights into why the model tends to use named entities more frequently than standard, more common words? Is the model possibly relying on named entities due to their higher specificity, allowing for greater diversity and (possibly) orthogonality compared to other words, thus enabling more detailed descriptions?

2) There’s a typo on line 163, an extra "is".

3) Do the authors plan on releasing the code upon acceptance?

**Limitations:**

The authors adequately addressed the limitations.

---

> ### Author Rebuttal · Authors · 2023-08-09
>
> Thank you for your review. We address your concerns and questions below.
>
> > Are results statistically significant with one run?
>
> We have conducted multiple runs on the few-shot prompting experiments. Across three independent runs with temperature = 0, the average scores are 29.039, 29.038, 29.039, indicating the evaluation is reliable with very low statistical variance. Training LQAE should be consistent across runs on such large datasets, due to time constraints, we were not able to finish multiple runs of LQAE training. However, we plan to incorporate such results in the final version.
>
> >  Do the authors have any insights into why the model tends to use named entities more frequently than standard, more common words?
>
> As the reviewer has hypothesized, we do believe that the tendency to use named entities is indeed to the long-tailed nature of the subset of the named entity vocabulary - to access a greater number of and more diverse codebook vectors for better reconstruction.
>
> > Do the authors plan on releasing the code upon acceptance?
>
> Yes, we plan to release all code on acceptance.

---

> > ### Comment · Reviewer_Z64R · 2023-08-13
> >
> > I want to thank the authors for the rebuttal. It addressed the comments I've raised in my review.
> >
> > Considering that and the other reviews/author responses, I confirm my acceptance recommendation.

---

> > > ### Author Response · Authors · 2023-08-19
> > > **Thank you for your response**
> > >
> > > We would like to thank the reviewer for their insightful response to our rebuttal, and recommending acceptance. If the reviewer intended to raise their score to 7, we would kindly ask the reviewer to update the score in the original review.

---

### Official Review · Reviewer_FHFJ · 2023-07-07

**Soundness:** 3 good
**Presentation:** 3 good
**Contribution:** 3 good
**Rating:** 6
**Confidence:** 4

**Summary:**

The proposed approach, LQAE, addresses the lack of visual grounding in large language models by aligning text and image data in an unsupervised manner. It encodes images as sequences of text tokens by quantizing image embeddings using a language codebook and reconstructing the original input with a masked version of the quantized embeddings and BERT. LQAE enables few-shot multi-modal learning, outperforming baseline methods in tasks like image classification and visual question answering with as few as 1-10 image-text pairs.

**Strengths:**

1. The paper proposes a simple but effective method to align text and image representation with pretrained models.
2. The proposed method allows image to be used in the same way as text, which could be adopted for downstream tasks.
3. The method doesn't require image-text pairs by unidireictionally align the image to text embeddings.

**Weaknesses:**

1. The proposed method may lack explanability compared to methods that use contrastive learning to train on text-image pairs.
2. I'm concerned about the technical novelty as there have been a lot of works trying to align and mix visual and text tokens, including BEiT, Parti, etc.
3. The trained representation cannot transfer between models, e.g., between RoBERTa and BERT.
4. The downstream tasks are focused on few-shot settings. How does LQAE perform when doing text-image fine-tuning?

**Questions:**

See weaknesses.

**Limitations:**

The authors have discussed limitations in the paper.

---

> ### Author Rebuttal · Authors · 2023-08-09
>
> Thank you for your review. We address your concerns below.
>
> > The proposed method may lack explainability compared to methods that use contrastive learning to train on text-image pairs.
>
> Interpretability is indeed an existing flaw in the method, as we do not use text-image pairs. However, we believe that it is a promising direction in future work to potentially train LQAE models with mixed unpaired and paired data for more sample (paired data) efficient learning than existing methods. In addition, LQAE shows promise for a more general framework of leveraging single-modality denoisers to align arbitrary multi-modal pairs, and can be useful in scenarios where unpaired data is much easier to acquire than paired data.
>
> > Technical novelty over prior works that try to align and mix visual and text tokens and does text-image finetuning, including BEiT, Parti, etc.
>
> We would like to emphasize that the novelty of our work over prior methods centers around enabling vision-language capabilities without requiring text-image pairs - in contrast to nearly every prior vision-language work (Frozen, Flamingo, GPT-4, Parti, BLIP, etc.). In addition, our method does not require fine-tuning, or backpropagating through an LLM, which can be expensive (e.g. 175B LLM), or impossible (no weights available and only through API / prompting). Although BeIT does not require text-image pairs, the method is unimodal in images, and does not contain any language information, unlike LQAE.
>
> > The trained representation cannot transfer between models, e.g., between RoBERTa and BERT.
>
> Yes, this is a limitation regarding our model. However, to the best of our knowledge  we do not believe that this is a common property of most vision-language models, such as Frozen or GPT-4, where both vision and language representations are model / finetuning specific.

---

> > ### Comment · Reviewer_FHFJ · 2023-08-18
> >
> > Thanks for your response. I've carefully read it and decide to keep my original recommendation.

---

> > > ### Author Response · Authors · 2023-08-19
> > > **Thank you for your response**
> > >
> > > We would like to thank the reviewer for detailed and positive assessment of our work. We found the reviewer’s questions and suggestions insightful and we thank the reviewer for responding to our rebuttal.

---

### Decision · Program_Chairs · 2023-09-21

**Decision:**

Accept (poster)

**Comment:**

This paper proposes LQAE for unsupervised image-text alignment. The central concept lies in the use of a VQ-VAE framework, replacing the conventional codebook with frozen token embeddings extracted from a language model. The authors have done a nice job of rebuttal. After rebuttal, it received scores of 5666, ranging from borderline accept to weak accept.

Reviewers are generally happy about the paper, commenting that (1) the proposed idea is interesting and novel; (2) the experiments are comprehensive with detailed ablation study; (3) the paper is well written and presented. The additional results on MiniGPT-4 etc. makes the evaluation more complete. On the other side, reviewers also commented that (1) the idea of converting vision features into text tokens seems not very natural, and this process will lost a lot of visual information. (2) The proposed method lacks explainability, and by the end, web-crawled image-text data is not that hard to acquire, weakening the motivation of the paper.

Overall, the proposed idea in this paper looks interesting and novel, and experiments are solid. On balance, the AC thinks that the merits of the paper outweigh the flaws, and therefore, would like to recommend acceptance of the paper.